# Human foveal cone photoreceptor topography and its dependence on eye length

Yiyi Wang[1], Nicolas Bensaid[2], Pavan Tiruveedhula[1,3], Jianqiang Ma[4], Sowmya Ravikumar[1,3], Austin Roorda[1,3]*

[1]School of Optometry, University of California, Berkeley, Berkeley, United States; [2]Carl Zeiss Meditec AG, Berlin, Germany; [3]Vision Science Graduate Group, University of California, Berkeley, Berkeley, United States; [4]Department of Mechanical Engineering, Ningbo University, Ningbo, China

**Abstract** We provide the first measures of foveal cone density as a function of axial length in living eyes and discuss the physical and visual implications of our findings. We used a new generation Adaptive Optics Scanning Laser Ophthalmoscope to image cones at and near the fovea in 28 eyes of 16 subjects. Cone density and other metrics were computed in units of visual angle and linear retinal units. The foveal cone mosaic in longer eyes is expanded at the fovea, but not in proportion to eye length. Despite retinal stretching (decrease in cones/mm$^2$), myopes generally have a higher angular sampling density (increase in cones/deg$^2$) in and around the fovea compared to emmetropes, offering the potential for better visual acuity. Reports of deficits in best-corrected foveal vision in myopes compared to emmetropes cannot be explained by increased spacing between photoreceptors caused by retinal stretching during myopic progression.
DOI: https://doi.org/10.7554/eLife.47148.001

*For correspondence: aroorda@berkeley.edu

## Introduction

There has been a rapid increase in prevalence of myopia, of all magnitudes, in the period between 1971–1972 and 1999–2004 (*Vitale, 2009*). Across sub-populations grouped by race, ethnicity and gender, several studies report axial length of the eye to be the primary variable related to myopia (*González Blanco et al., 2008*; *He et al., 2015*; *Iyamu et al., 2011*). Increased axial length is associated with retinal stretching and thinning of posterior segment layers and the choroid (*Fujiwara et al., 2009*; *Harb et al., 2015*) and is associated with sight-threatening, often irreversible pathologies of the retina (*Morgan et al., 2012*; *Verkicharla et al., 2015*). Even without any detectable pathology, the structural changes associated with eye growth ought to have functional consequences for vision.

### What do we know about functional deficits in myopia?

One might expect that eye growth would stretch the photoreceptor layer and would increase the spacing between cones, causing a longer eye to more coarsely sample an image relative to a shorter eye. However the situation is not that simple; the axial elongation associated with eye growth is accompanied by magnification of the retinal image (*Strang et al., 1998*). If the enlargement of the retinal image exactly matched the stretching of the cone mosaic, then eyes of different lengths would sample the visual field similarly. In fact, in large scale studies, myopes generally attain reasonably good visual acuity with optical correction (*He et al., 2004*; *Jong et al., 2018*).

**eLife digest** The human eye has many different parts that enable sight. The retina is the light sensitive tissue at the back of the eye, and it contains the fovea, the region that provides the clearest vision. Light must be focused on the retina to create images, a feat achieved by the transparent parts at the front of the eye called the cornea and lens. These parts of the eye are called the optics.

Between birth and adulthood, significant changes take place in the eye. Most noticeably, the distance between the optics and the fovea grows by about seven millimeters. Cells called cone photoreceptors, which provide light sensitivity, migrate and pack into the fovea. Finally, the eye's optics adjust to maintain a sharp focus. At the same time, the brain is learning how to process inputs from the eyes to generate mental images that realistically correspond to the physical world around it. The development of the eye is fascinating in its complexity, but for more and more people, the process does not go as expected.

Specifically, a growing proportion of the population has eyes that are too long. This means that, for light reflected by far away objects, the eye's optics form an image in front of the retina instead of on it. As a result, images of distant objects cannot be seen clearly, a condition known as myopia or nearsightedness. Researchers have also discovered that nearsighted people see less clearly than those who do not use glasses, even when given a sharp image to examine at close range. It has been hypothesized that these deficits result from stretching of the retina as the eye becomes bigger.

Until recently, testing this hypothesis by looking at cone photoreceptors directly in the eye was impossible. This is because the optics of all eyes have small imperfections that distort the light passing through them, including any light used to take high resolution microscopic images of the fovea. This hurdle can be overcome using adaptive optics, which means adding a deformable mirror to the instrument being used to image the eye that can adjust to correct the distortion.

Wang et al. use a new generation Adaptive Optics Scanning Laser Ophthalmoscope to check the density of cones at the fovea in relation to the size of the eye. They show that although the center of the fovea has fewer cones when the eye is bigger, this effect is more than offset because the longer eye increases magnification. So, if a near-sighted person wearing contacts and someone who does not need glasses stood side-by-side admiring the full moon, the near-sighted person would most likely have more cones sampling the image and should therefore have a higher resolution view.

These findings rule out reductions in the density of cone photoreceptors as the cause or effect of visual deficits associated with near-sightedness, adding to the understanding of this common condition.

DOI: https://doi.org/10.7554/eLife.47148.002

However, more careful inspection reveals that myopes generally (6 out of 9 studies) have poorer angular resolution and have uniformly (3 out of 3 studies) poorer retinal resolution. *Table 1* summarizes published results from psychophysical foveal tasks.

Most notably, *Atchison et al. (2006)* and *Coletta and Watson (2006)* show clear deficits in retinal resolution (cyc/mm) with increasing myopia using interferometric methods which bypass the optics of the eye and *Rossi et al. (2007)* show significant deficits in angular resolution (cyc/deg) in low myopes, even after using adaptive optics to correct for optical blur. All studies that find myopic visual deficits implicate retinal stretching as a possible cause, but what is actually happening structurally at the foveal center during myopic progression is not known. Therefore, the aim of the current study is to more carefully investigate how the length of the eye affects cone density at and near the foveal center.

## Models for how photoreceptors change with eye growth

Two types of cone densities will be discussed in this study. Linear density quantifies how many cones are within a fixed area, in square mm, and serves as a way to evaluate physical retinal stretching caused by eye growth. Angular density quantifies how many cones are within one degree visual angle, (the visual angle on the retina is measured from the secondary nodal point of the eye).

Angular density serves as a way to evaluate the visual implications of eye growth as it governs the sampling resolution of the eye.

*Figure 1* illustrates three models, along the lines of *Strang et al. (1998)*, of how photoreceptor structure might be affected by myopic eye growth. In the first model, called the global expansion model, the retina is proportionally stretched with increasing axial length - cones are more spaced out in longer eyes - and linear density decreases with eye length. Assuming that the secondary nodal point remains at a fixed position relative to the anterior segment, the number of cones within a fixed angular area will remain constant. Therefore, angular cone density will be constant with eye length.

**Table 1.** Summary of studies investigating foveal spatial vision and sensitivity tasks in myopia.

| Author | Refractive error range of myopic cohort [D] | Functional tests | Results for myopes at foveal center | Suggested cause |
|---|---|---|---|---|
| *Fiorentini and Maffei, 1976* | −5.5 to −10 (n = 10) | CSF | Reduced CSF | Neural insensitivity (myopic amblyopia) |
| *Thorn et al., 1986* | −6 to −9.75 (n = 13) | CSF | No difference in CSF | Global expansion |
| *Collins and Carney, 1990* | −2 to −11 (n = 16) | VA, CSF | No difference in VA or CSF between low and high myopic groups with contact lens correction | NA |
| *Strang et al., 1998* | 0 to −14 (n = 34) | VA | Reduced VA (MAR) with increasing myopia after controlling for spectacle magnification | Retinal expansion specifically at the posterior pole; increased aberrations |
| *Liou and Chiu, 2001* | 0 to >-12 (n = 105 eyes) | CSF | Reduced CSF with increasing myopia | Retinal stretching and disruption, neural insensitivity (myopic amblyopia) |
| *Chui et al., 2005* | −0.5 to −14 (n = 60) | Grating resolution | Decreased resolution acuity in cyc/mm | Retinal expansion specifically at the posterior pole; global expansion along with ganglion cell loss |
| *Coletta and Watson, 2006* | +2 to −15 (n = 17) | Interferometric grating resolution | Decreased resolution acuity in cyc/mm but not in cyc/deg | Retinal expansion specifically at the posterior pole |
| *Atchison et al., 2006* | +0.75 to −12.4 (n = 121) | Spatial summation; interferometric grating resolution | Increased critical summation area in linear area, but not in angular area; Decreased resolution acuity in cyc/mm but not in cyc/deg | Retinal expansion specifically at the posterior pole; global expansion along with ganglion cell loss |
| *Stoimenova, 2007* | −1 to −8 (n = 60) | Contrast thresholds of 20/120 letters | Lower sensitivity to contrast for letters with a fixed angular size | Morphologic changes in the retina |
| *Rossi et al., 2007* | −0.5 to −3.75 (n = 10) | AO-corrected VA | Reduced acuity (MAR) compared to emmetropes | Retinal expansion, neural insensitivity; neural insensitivity (myopic amblyopia) |
| *Jaworski et al., 2006* | −8.5 to −11.5 (n = 10) | Foveal summation thresholds; CSF | Increased critical summation area (angular) Decreased luminance sensitivity Reduced contrast sensitivity at high frequencies (cyc/deg) | Reduction in photoreceptor sensitivity; postreceptoral changes; increased aberrations |
| *Ehsaei et al., 2013* | −2.00 to −9.62 (n = 60) | Size threshold of high and low contrast letter targets | No difference in threshold retinal image size between myopes and emmetropes. | NA |

DOI: https://doi.org/10.7554/eLife.47148.003

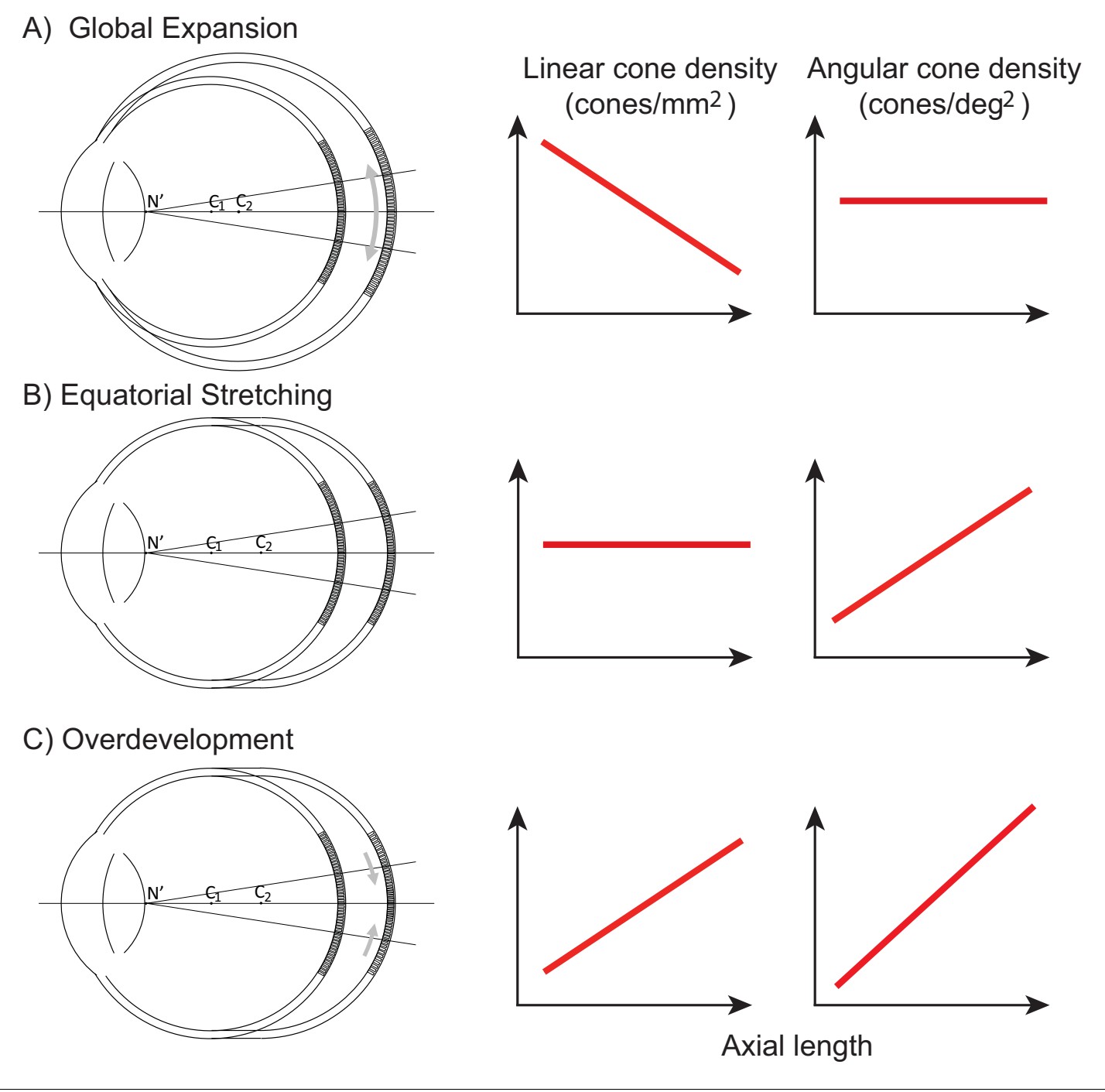

**Figure 1.** Three models of myopic eye growth. (A) Global expansion shows an eyeball that is proportionally stretched. (B) The equatorial stretching model indicates a growth model where the fovea stays rigid and unaffected as the eye grows. (C) The over-development model shows that myopic eye growth is similar with developmental eye growth where photoreceptors continue to migrate towards the fovea as the eye grows.

DOI: https://doi.org/10.7554/eLife.47148.004

In the second model, called the equatorial stretching model, the posterior retina simply moves axially further from the anterior segment of the eye so that the linear density does not change with eye length. Since the retina is moving further from the secondary nodal point, more cones will fall within a fixed angular area and the angular cone density will increase with eye length. The final model,

called the over-development model, describes a structural photoreceptor change that mimics the changes that occur during development (*Springer and Hendrickson, 2004*) whereby the photoreceptors continue to migrate towards the fovea as the eye grows. In this scenario, longer eyes will show both increased linear cone density and an even steeper increase in angular cone density. The model is motivated by observations of increased linear cone density in the foveas of marmosets that underwent lens-induced eye growth (*Troilo, 1998*).

## Previous studies of cone spacing with axial length

The most definitive studies of cone spacing as a function of axial length are done through direct imaging of the retina – wherein sharp images of the cones are enabled through the use of adaptive optics, a set of technologies that actively compensate the blur caused by aberrations of the eye (*Liang et al., 1997*). Combined with confocal scanning laser ophthalmoscopy (*Webb et al., 1987*), adaptive optics offers the highest contrast *en face* images of the foveal photoreceptor mosaic ever recorded in vivo (*Dubra et al., 2011*; *Roorda et al., 2002*).

Despite continued advances in image quality, previous studies investigating cone packing and eye length have not made their measurements at the foveal center, the most important region for spatial vision but the most difficult to image owing to the small size of photoreceptors. There are a number of studies on cone packing and eye length (*Chui et al., 2008*; *Elsner et al., 2017*; *Kitaguchi et al., 2007*; *Li et al., 2010*; *Obata and Yanagi, 2014*; *Park et al., 2013*) and here we summarize the published results that are most relevant to our study. *Chui et al. (2008)* investigated angular and linear cone density at 1 mm and three degrees eccentricity. They found a significant decrease (p<0.05) in linear cone density as a function of eye length at 1 mm (which, by angular distance, is closer to the fovea in a longer eye than in a shorter eye) in all directions except in the nasal retina. They found that the angular cone density at three degrees (which, by linear distance, is closer to the fovea, in a shorter eye than in a longer eye) increased with eye length, but the trends were not significant. *Li et al. (2010)* made similar measures, but closer to the fovea (from 0.10 mm to 0.30 mm eccentricity). They found that linear cone density decreased with eye length, but the trends were not significant at the smallest eccentricities (0.1 and 0.2 mm). When the data were plotted in angular units and angular distance from the fovea, they found that angular cone density trended toward an increase with eye length but none of the trends were significant. A more recent study measured peak cone densities in the fovea as well as axial length for 22 eyes of 22 subjects (*Wilk et al., 2017*) but they did not plot peak cone density as a function of axial length, as it was not the aim of their study. We plotted the data they provided in their paper and found that the linear cone density at the foveal center dropped significantly with increases in axial length, similar to what was found by *Li et al. (2010)* and *Chui et al. (2008)*, but the angular cone density had no dependency on eye length. Summary plots from previous literature are shown in *Figure 2AB*.

*Wilk et al. (2017)*'s data were consistent with a global expansion model and *Li et al. (2010)* and *Chui et al. (2008)*'s data only leaned toward a model that falls between the global expansion and equatorial stretching models. If the trends found by *Li et al. (2010)* and *Chui et al. (2008)* near the fovea were to extend to the foveal center, then myopes would have higher foveal photoreceptor sampling resolution with a consequent potential for better performance on visual tasks compared to emmetropes. As such, the simplest explanation for visual deficits in myopes – increased separation between cones caused by retinal stretching – would have to be ruled out.

With the improvements in resolution of adaptive optics ophthalmoscopes, imaging the smallest cones at the foveal center is now possible in many eyes, enabling a definitive analysis of the cone density at the fovea as a function of eye length.

## Results

The experiments were approved by the University of California, Berkeley Committee for the Protection of Human Subjects. All subjects provided informed consent prior to any experimental procedures. Subjects self-reported their eye health so that only healthy individuals with no ocular conditions were included in the study. All eyes were dilated and cyclopleged with 1% Tropicamide and 2.5% Phenylephrine before imaging. We report data from 28 eyes of 16 subjects with a wide range of refractive error and axial length. Age, sex and ethnicity are listed on *Table 2*.

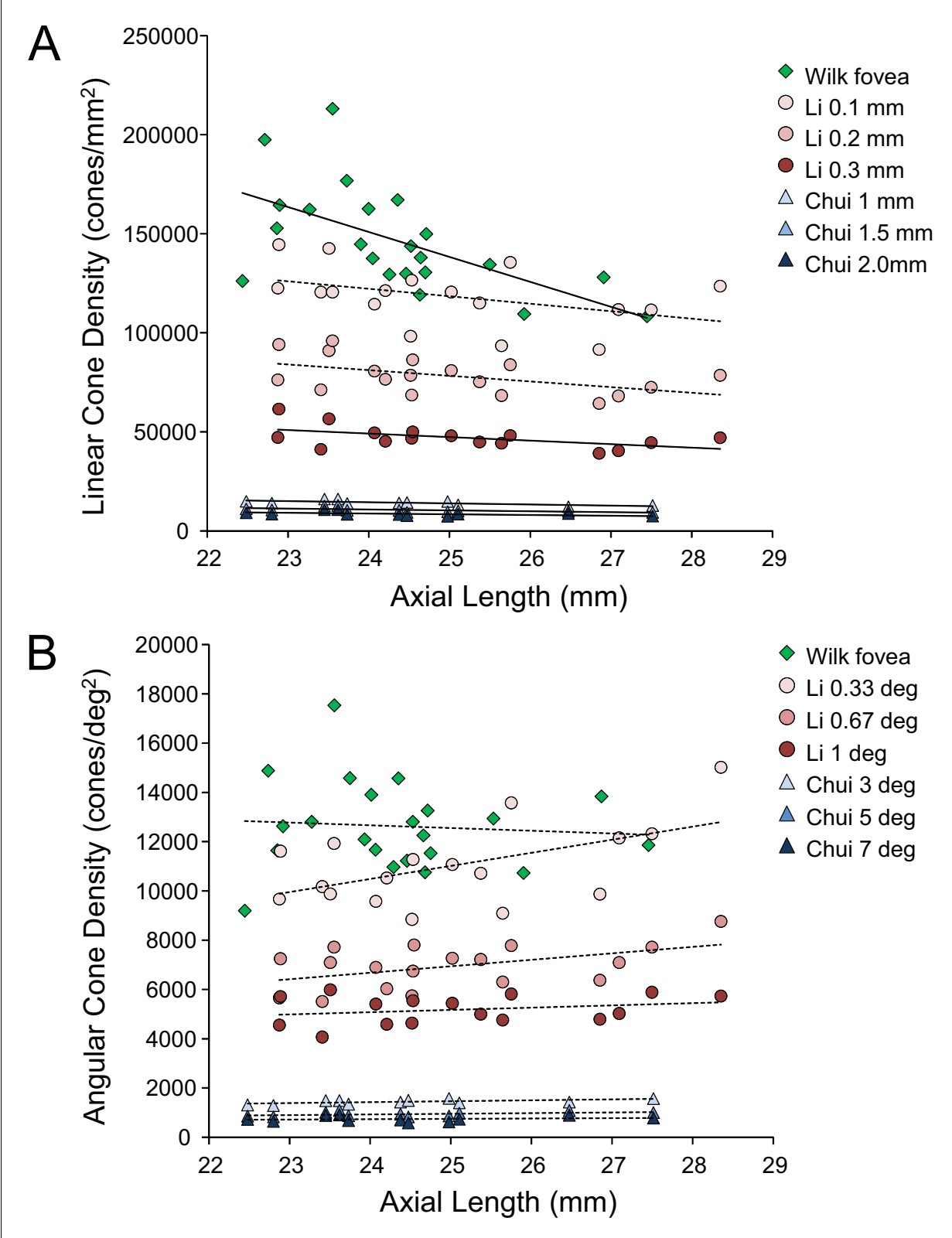

**Figure 2.** Summary of published data from *Li et al. (2010)*, *Chui et al. (2008)* and *Wilk et al. (2017)*. In both plots, the linear fits with the solid lines indicate the data that have significant trends. (**A**) Linear cone density has a decreasing trend with axial length near the fovea. (**B**) Angular cone density (sampling resolution) of the eye generally increases with axial length although none of the data show a significant linear relationship.
DOI: https://doi.org/10.7554/eLife.47148.005

**Table 2.** Subject details, biometry and cone density for all subjects.
Each subject's refractive error was self-reported at the time of the study. Axial Length, corneal curvature and anterior chamber depth were measure by IOL Master, and retinal magnification factor (microns/deg) was calculated from biometry data. Linear and angular cone densities are reported for a 10 arcminute sampling window (see Materials and methods).

| Subject ID | Eye | Gender | Age | Ethnicity | Spherical equivalent refraction (D) | Axial length (mm) | Corneal curvature (mm) | Anterior chamber depth (mm) | Retinal magnification factor (microns/deg) | Angular cone density (cones/deg²) | Linear cone density (cones/mm²) | PRL distance from fovea (minutes) | PRL distance from fovea (microns) | PRL angular cone density (cones/deg²) | PRL linear cone density (cones/mm²) |
|---|---|---|---|---|---|---|---|---|---|---|---|---|---|---|---|
| 20165 | L | F | 28 | Caucasian | 0.500 | 22.26 | 7.37 | 3.86 | 261.79 | 13316 | 194625 | 2.83 | 12.34 | 12470 | 181952 |
| | R | F | 28 | Caucasian | 0.500 | 22.64 | 7.44 | 3.80 | 267.79 | 12714 | 177692 | 5.30 | 23.66 | 11758 | 163965 |
| 20177 | L | F | 18 | Mixed | 0.000 | 23.04 | 7.80 | 3.24 | 273.59 | 12211 | 162890 | 7.25 | 33.06 | 11476 | 153319 |
| | R | F | 18 | Mixed | 0.000 | 23.23 | 7.91 | 3.20 | 275.85 | 11999 | 159027 | 4.57 | 21.00 | 11317 | 148721 |
| 10003 | L | M | 50 | Caucasian | 1.000 | 23.30 | 7.80 | 3.12 | 278.81 | 15851 | 204020 | 7.23 | 33.59 | 13961 | 179594 |
| | R | M | 50 | Caucasian | 1.000 | 23.50 | 7.81 | 3.14 | 282.00 | 15358 | 193090 | 6.45 | 30.32 | 14869 | 186972 |
| 20176 | L | F | 18 | Asian | 0.000 | 23.45 | 7.98 | 3.65 | 276.50 | 12515 | 163676 | 18.16 | 83.71 | 8813 | 115273 |
| | R | F | 18 | Asian | 0.000 | 23.58 | 8.01 | 3.62 | 278.52 | 12312 | 158356 | 4.05 | 18.78 | 11913 | 153566 |
| 20172 | L | F | 25 | Caucasian | −0.750 | 23.56 | 7.71 | 3.90 | 280.13 | 15516 | 196844 | 1.23 | 5.72 | 15210 | 193824 |
| | R | F | 25 | Caucasian | −0.500 | 23.65 | 7.72 | 3.96 | 281.33 | 14976 | 189377 | 3.13 | 14.66 | 14636 | 184921 |
| 20147 | R | M | 26 | Caucasian | −0.375 | 24.16 | 7.73 | 2.36 | 298.73 | 15537 | 174122 | 4.68 | 23.29 | 14839 | 166278 |
| | L | M | 26 | Caucasian | 0.000 | 24.17 | 7.81 | 4.03 | 288.94 | 14994 | 178435 | 11.57 | 55.72 | 13894 | 166422 |
| 20124 | L | F | 26 | Asian | −3.000 | 24.67 | 7.70 | 4.05 | 298.82 | 13973 | 153998 | 5.17 | 25.77 | 13334 | 149334 |
| | R | F | 26 | Asian | −4.250 | 25.29 | 7.68 | 4.07 | 309.88 | 13927 | 145588 | 2.38 | 12.30 | 13543 | 141033 |
| 20174 | L | F | 43 | Caucasian | −1.750 | 24.80 | 7.79 | 3.57 | 302.57 | 13775 | 150204 | 7.78 | 39.21 | 11671 | 127480 |
| | R | F | 43 | Caucasian | −2.750 | 25.37 | 7.83 | 3.62 | 311.85 | 12857 | 132443 | 6.00 | 31.19 | 11848 | 121826 |
| 20173 | R | F | 22 | Caucasian | −2.750 | 24.96 | 7.81 | 3.68 | 304.64 | 16648 | 179779 | 7.11 | 36.08 | 15989 | 172286 |
| 20170 | R | M | 26 | Asian | −2.250 | 25.00 | 7.69 | 3.90 | 305.54 | 14485 | 153681 | 8.98 | 45.73 | 12244 | 131153 |
| | L | M | 26 | Asian | −3.750 | 25.66 | 7.65 | 4.15 | 316.25 | 14853 | 147115 | 1.70 | 8.96 | 14708 | 147060 |
| 20138 | R | F | 29 | Caucasian | −5.000 | 25.26 | 7.95 | 3.14 | 311.22 | 13874 | 141971 | 6.32 | 32.76 | 12449 | 128530 |
| | L | F | 29 | Caucasian | −5.000 | 25.28 | 7.91 | 3.15 | 311.92 | 14776 | 151699 | 5.36 | 27.87 | 14060 | 144506 |
| 20114 | R | F | 24 | Asian | −5.500 | 25.83 | 8.72 | 3.47 | 310.94 | 14615 | 152657 | 7.00 | 36.29 | 13787 | 142601 |
| | L | F | 24 | Asian | −6.000 | 26.16 | 8.98 | 3.58 | 313.31 | 15634 | 159228 | 4.48 | 23.40 | 15287 | 155729 |
| 20160 | R | F | 25 | Asian | −5.375 | 25.83 | 7.81 | 3.60 | 320.25 | 15885 | 155083 | 8.25 | 44.06 | 14409 | 140492 |
| 20143 | R | F | 23 | Asian | −6.875 | 25.91 | 7.42 | 2.10 | 334.12 | 17258 | 153560 | 3.01 | 16.77 | 16562 | 148354 |
| 20158 | R | F | 34 | Asian | −6.500 | 26.60 | 7.84 | 3.51 | 333.78 | 13147 | 118491 | 11.82 | 65.76 | 10876 | 97623 |
| 20163 | R | F | 25 | Asian | −7.125 | 26.84 | 7.89 | 3.65 | 336.60 | 18114 | 159397 | 3.82 | 21.42 | 17481 | 154287 |
| | L | F | 25 | Asian | −7.125 | 27.06 | 7.89 | 3.65 | 340.44 | 19001 | 163731 | 5.02 | 28.50 | 17899 | 154437 |

DOI: https://doi.org/10.7554/eLife.47148.006

## Biometry data

All the biometric measures used to convert angular dimensions to linear retinal dimensions are listed on *Table 2*. The strong correlation between refractive error and eye length (p<0.0001) indicates that the myopia in this cohort was predominantly as a result of axial length.

## Imaging data

Images of the foveal region, the preferred retinal locus for fixation (PRL) and the fixation stability were recorded with an adaptive optics scanning laser ophthalmoscope (see Materials and methods). The image of one subject (10003L) is shown in *Figure 3A*. All the cones were resolved with our imaging system. The scatter plot indicates the scatter plot of fixation over the course of a 10 s video. *Figure 3B* shows the same image with all cones labeled and a color-coded overlay indicating the density. 16,184 labeled cones are shown on the figure. The point of maximum density is indicated by the black cross and the PRL is indicated by the yellow ellipse (best fit to the scatter plot locations in *Figure 3A*). This eye has a peak linear density of 204,020 cones/mm$^2$, and a peak angular density of 15,851 cones/deg$^2$. Cone density plots in linear and angular units for all eyes are shown on *Figure 3—figure supplements 1* and *2*. Original images and a list of the cone locations for each can be downloaded from the Dryad Digital Repository https://datadryad.org/review?doi=doi:10.5061/dryad.nh0fp1b.

*Figure 4* shows the linear cone density as a function of linear eccentricity, where the average linear cone density was computed in 25-micron wide annuli centered around the point of peak density.

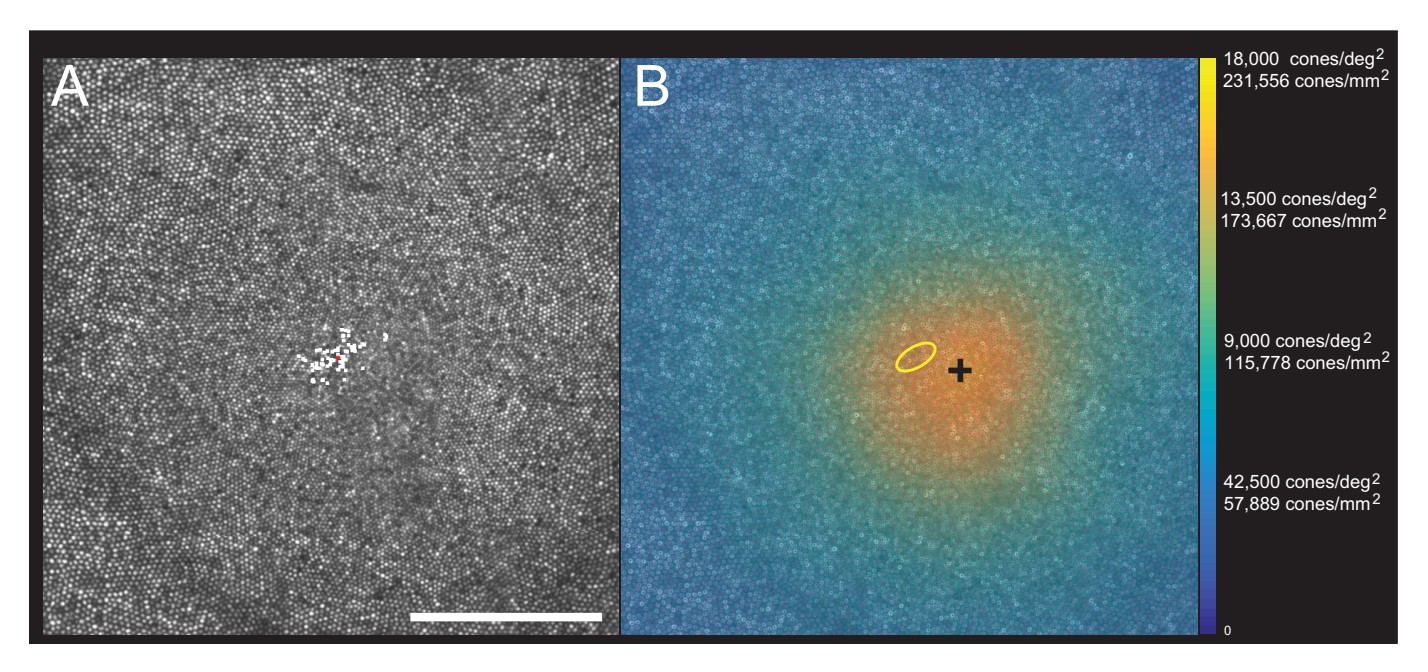

**Figure 3.** Image, PRL, cone locations and density plot for one subject. (**A**) AOSLO image of the fovea of subject 10003L. Only the central 1.5 degrees are shown here (810 × 810 pixels), which contains 16,184 cones. The white dots are a scatter plot showing the PRL, or position of the fixated stimulus over the course of a 10 s video. The red dot is the centroid of the scatter plot. (**B**) Same image with a color overlay indicating the density. Linear and angular cone densities are indicated on the right colorbar. Peak cone densities in this eye are 204,020 cones/mm$^2$ and 15,851 cones/deg$^2$. The yellow ellipse is the best fitting ellipse containing ~68% of the points in the scatterplot and indicates the PRL. The black cross indicates the position of peak cone density. Scale bar is 0.5 degrees, which in this eye corresponds to 139.4 microns.

DOI: https://doi.org/10.7554/eLife.47148.007

The following figure supplements are available for figure 3:

**Figure supplement 1.** Linear cone density (cones/mm$^2$) plots over the central 450 microns for all 28 eyes.
DOI: https://doi.org/10.7554/eLife.47148.008
**Figure supplement 2.** Angular cone density (cones/deg$^2$) plots over the central 1.5 degrees for all 28 eyes.
DOI: https://doi.org/10.7554/eLife.47148.009

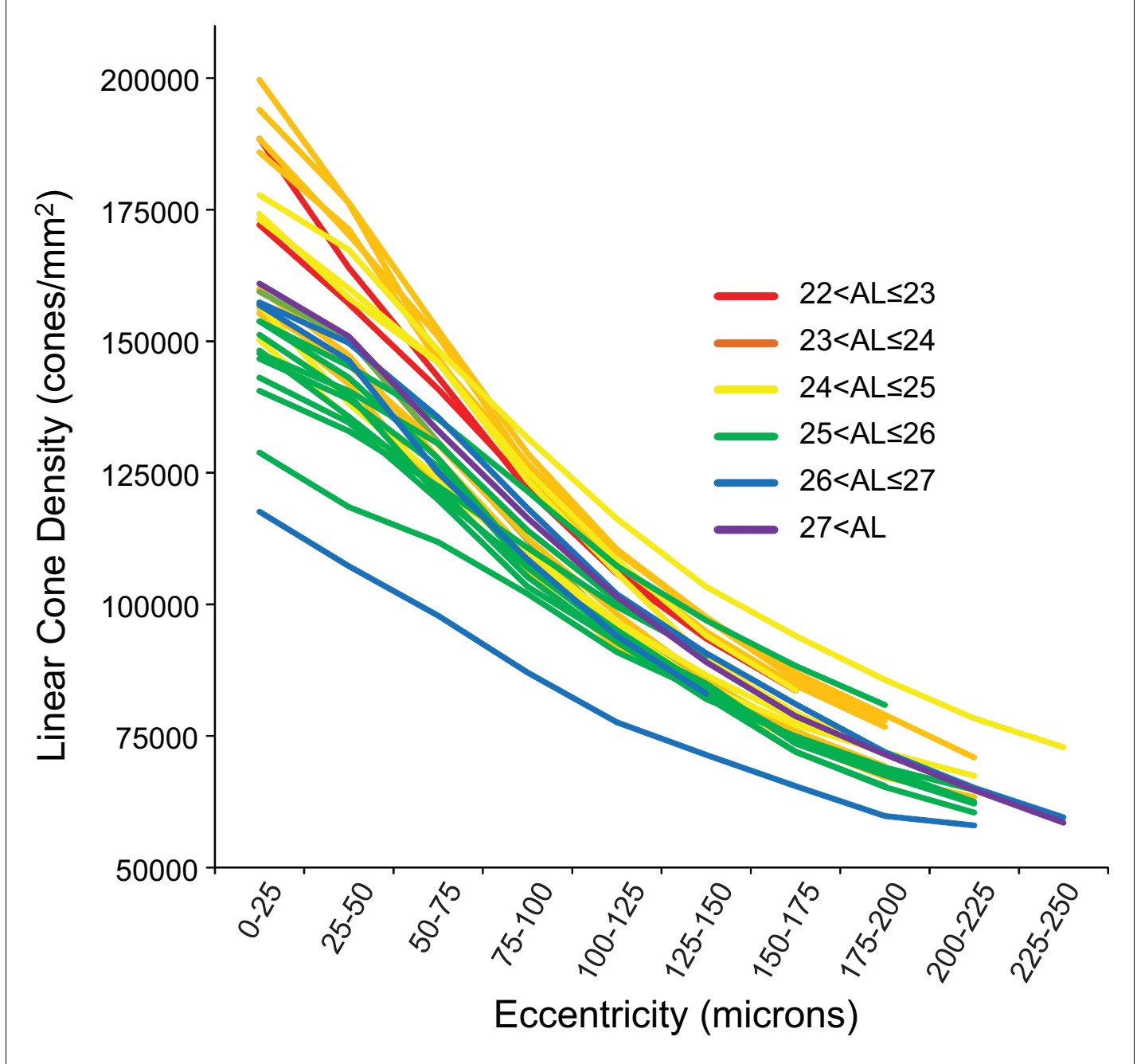

**Figure 4.** Cone density as a function of eccentricity for all eyes. The axial length ranges of the subjects are color coded, with warmer colors for shorter eyes and cooler colors for longer eyes. In this plot, it is apparent that shorter eyes generally have higher peak cone densities.

DOI: https://doi.org/10.7554/eLife.47148.010

The following source data and figure supplements are available for figure 4:

**Source data 1.** Data for plots of cone density as a function of eccentricity for all subjects.
DOI: https://doi.org/10.7554/eLife.47148.015

**Figure supplement 1.** Plots of average cone density of all 28 eyes as a function of eccentricity in units of.
DOI: https://doi.org/10.7554/eLife.47148.011

**Figure supplement 1—source data 1.** Data for plots of average linear and angular cone density as a function of eccentricity.
DOI: https://doi.org/10.7554/eLife.47148.012

**Figure supplement 2.** Plots of density as a function of eccentricity in the vertical and horizontal directions.
DOI: https://doi.org/10.7554/eLife.47148.013

*Figure 4 continued on next page*

*Figure 4 continued*

**Figure supplement 2—source data 1.** Data for plots of average linear and angular cone density as a function of eccentricity in the horizontal and vertical directions.

DOI: https://doi.org/10.7554/eLife.47148.014

*Figure 4—figure supplement 1* are plots of the average cone densities in linear and angular units as a function of eccentricity in microns and arcminutes. *Figure 4—figure supplement 2* are plots of the average linear and angular cone densities as a function of horizontal and vertical direction.

In order to show the trends of density with axial length *Figure 5A and B* plot linear and angular cone density as a function of axial length where the colors indicate different eccentricity - red to purple indicate distance from the from fovea towards more parafoveal locations. *Figure 5A* reveals that peak linear density decreases significantly with axial length and the trend persists and remains significant from the fovea out to 100 microns eccentricity. Axial length accounts for 39% of the variance in the changes in linear cone density. *Figure 5B* shows the opposite trends when plotted in angular units. Peak angular density increases significantly with axial length and the trend persists and remains significant out to 40 arcminutes eccentricity. Axial length accounts for 32% of the variance in the changes in angular cone density. The plots clearly indicate that although stretching does occur (*Figure 5A*) it is not a simple global expansion and longer eyes have higher sampling density. The trends hold at and around fovea with statistical significance.

A more relevant measure of the impact of eye length on vision is how the angular cone density changes at the PRL, which is often displaced from the location of peak cone density (*Li et al., 2010*; *Putnam et al., 2005*; *Wilk et al., 2017*). If, for example, longer eyes had more displaced PRLs then that could diminish, or even reverse, the trend of increased angular density with eye length reported in *Figure 5B*. We found that the average displacement between PRL and maximum cone density was 6.1 arcminutes and 30.4 microns. There was no significant linear relationship found between PRL displacement in either angular or linear units vs. axial length. Therefore, the PRL was not more displaced in myopes than in emmetropes from the point of peak cone density. Plots of the cone density at the PRL with axial length show the same trend at the PRL as at the point of maximum cone density (*Figure 6A and B*).

Finally, we explored whether fixational eye movements might have a dependency on axial length. Fixation stability around the PRL had an average standard deviation of 3.94 arcminutes and 19.84 microns. The average area of the best fitting ellipse containing ~68% of the points in the scatterplot (defined as the bivariate contour ellipse area, or BCEA) was 50.7 square arcminutes and 1303 square microns. The plot of BCEA in square microns vs. axial length vs. showed a trend that approached significance (p=0.0596) (*Figure 7A*), but when we plotted BCEA in square arcminutes vs. axial length, the trend was no longer apparent (p=0.364) (*Figure 7B*). In other words, if there is any increase in fixational eye movements in microns, it is just a symptom of having a longer eye.

## Discussion

In this paper, we measure the cone density at and near the foveal center and investigate how it changes as a function of axial length. This is the first comprehensive study of cones in living eyes at the foveal center, the area solely responsible for a human's fine spatial vision. Our results show that although some expansion does occur (linear cone density decreases with axial length) the angular sampling resolution actually increases, on average, with axial length. Prior to this study, the relationships between cone density and axial length were only made outside of the fovea, the closest being 0.1 mm, or 0.3 degrees (*Li et al., 2010*). Although an eccentricity of 0.3 degrees might seem close, it is noted that the cone density drops precipitously just outside of the location of peak density (*Curcio et al., 1990*) as does human vision (*Poletti et al., 2013*) (*Rossi and Roorda, 2010b*). There are other factors that govern peak cone density, however; eye length accounts for anywhere between 31% and 39% of the variance in cone density at the foveal center.

Our finding that the slopes of cone density vs. axial length are in opposite directions when plotted in linear (negative slope) and angular (positive slope) units, supports an eye growth model that lies between the global expansion model and an equatorial stretching model. Previous studies from our lab (*Li et al., 2010*) and also from *Chui et al. (2008)* leaned in the same direction. None of the

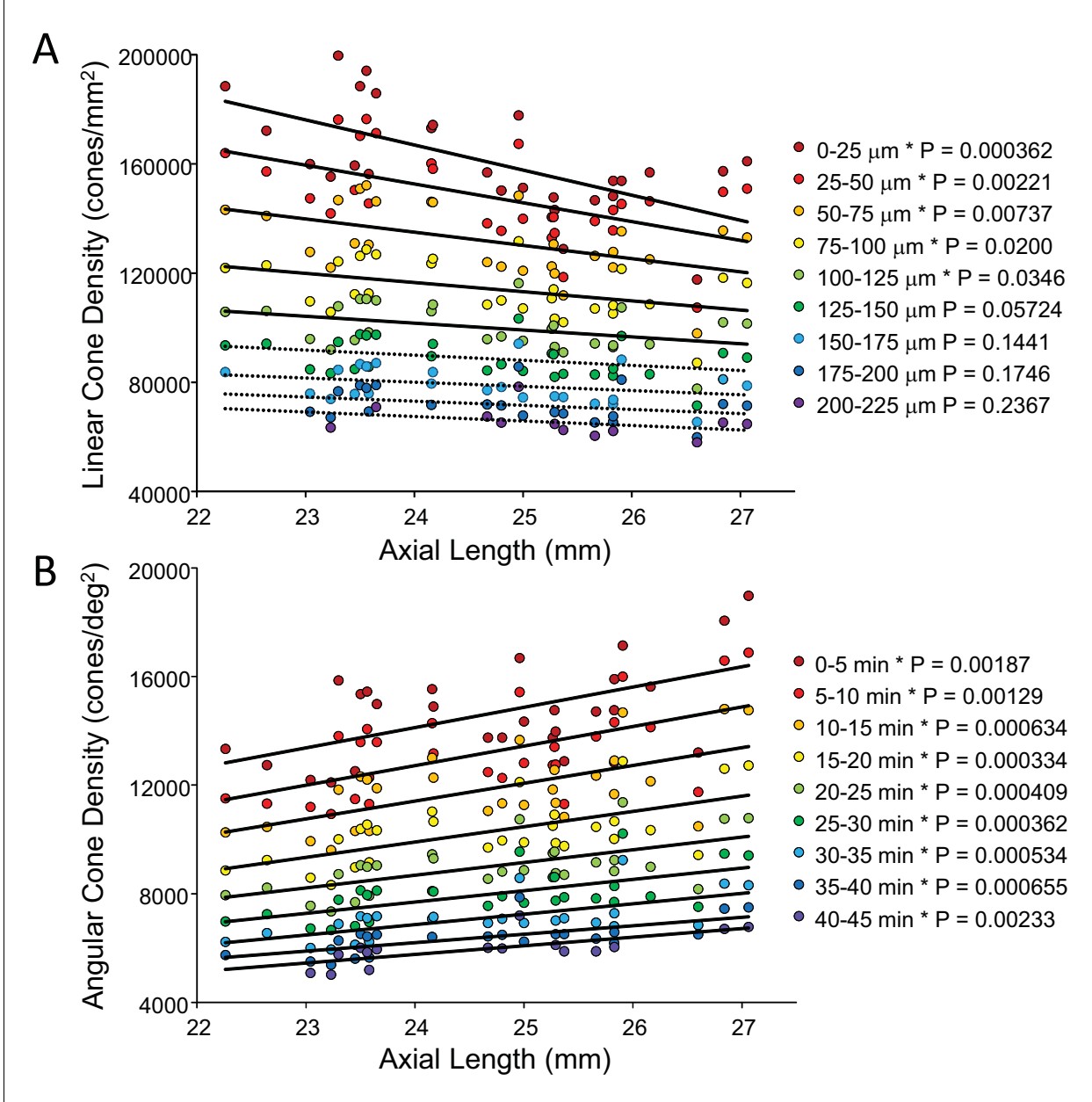

**Figure 5.** Plots of cone density as a function of axial length at and near the fovea. (**A**) Linear cone densities as a function of axial length. Longer eyes have lower linear cone density than shorter eyes. The trend remains significant out to 100 microns eccentricity. At the peak, the details for the trendline are: slope = −3,185 with 95% confidence intervals from −4,578 to −13,793. (**B**) Angular cone densities as a function of axial length. The peak angular cone density increases significantly with increasing axial length and this trend remains significant out to 40 arcminutes eccentricity. At the peak, the details for the trendline are: slope = 749 with 95% confidence intervals from 304 to 1193. Relationships with p-values<0.05 are labeled with asterisks and trendlines are shown as solid lines. Relationships with p-values≥0.05 have dashed trendlines.

DOI: https://doi.org/10.7554/eLife.47148.016

The following source data is available for figure 5:

**Source data 1.** Data for plots of cone density as a function of axial length at and near the fovea.

DOI: https://doi.org/10.7554/eLife.47148.017

cone density studies provide insight into the reasons why the photoreceptor density would behave this way with eye growth, but the results do align with other observations reported in the literature. Specifically, *Atchison et al. (2004)* used magnetic resonance imaging and found that eyeball

dimensions in axial myopes are variable but are generally larger in all directions with a weak tendency to be preferentially greater in the axial direction. Their reported eye growth patterns lie between that illustrated for the global expansion and equatorial stretching models in *Figure 1*.

Our results differ from *Wilk et al. (2017)* whose data support a global expansion model (i.e. there is no detectable change in angular cone density with axial length; *Figure 2B*). But it is important to point out that their study did not set out to address the same question and the number of subjects with long axial lengths was disproportionately low.

Our results also differ from *Troilo (1998)* who studied retinal cell topography in a marmoset animal myopia model. Higher cone packing densities were observed in the experimentally enlarged eyes compared to normal eyes in the fovea. Their result followed the overdevelopment model, which is the reason why we included it as one of the possible outcomes of our study. In fact, the overdevelopment model is an extension of Springer's model of development (*Springer and Hendrickson, 2004*), which offers a biomechanical explanation for how cone packing increases at the foveal center in a developing eye. While our data do not support the overdevelopment model, it does not preclude the existence of biomechanical factors working in opposition to simple global expansion.

The fact that angular cone density (visual sampling resolution) increases with eye length (myopia), at the peak density and at the PRL, means that poorer performance by myopes on resolution tasks cannot be explained by a decrease in photoreceptor sampling. The deficit must arise at a post-receptoral level.

Low-level causes for myopic visual deficits might arise from differences in the connectivity between cones and ganglion cells. *Atchison et al. (2006)* suggested that abnormal eye growth may be associated with a loss of ganglion cells. Alternately, if ganglion cells pool signals from multiple cones, then they will impose the retinal sampling limit and reduce certain aspects of visual performance (acuity, for example). Recent electron microscopy studies of a human fovea have revealed extensive convergence and divergence connections between photoreceptors and ganglion cells, albeit in an eye from an individual who was born prematurely (*Dacey, 2018*). These discoveries challenge our current understanding of neural connectivity in the foveal center and force us to consider the possibility of interindividual differences in foveal cone wiring. More experiments are necessary to explore these ideas.

To explain why low myopes did not perform as well on an acuity task as emmetropes, even after correction or bypassing of high order aberrations, *Rossi et al. (2007)* and *Coletta and Watson (2006)* both raised the possibility that myopes might have become desensitized to high frequency information (low level myopic amblyopia) as a result of having less exposure to a high contrast visual environment. In this case, it might be possible to train myopes to take advantage of their higher sampling resolution, but one myope in a follow up study by *Rossi and Roorda (2010a)* never reached the acuity levels of emmetropes in the same study.

## Comparisons with previous studies

### Peak cone densities

*Curcio et al. (1990)* measured spatial density of cones and rods in eight explanted whole-mounted human retinas. They found a large range of peak foveal cone densities with an average of 199,000 cones/mm$^2$. Their density measurements were made over a window size of 29 $\times$ 45 microns. Our range of peak cone densities measured over a 10 arcmin (~50 micron) sampling window was 118,491 to 204,020 with an average of 163,242 cones/mm$^2$. When we averaged the peak cone density over a circular aperture of 7.5 arcminutes which was more similar to the sampling window that *Curcio et al. (1990)* used to compute density, we measured peak linear cone densities ranging from 123,611 to 208,606 with an average of 166,854 cones/mm$^2$. See Materials and methods for more discussion on the implications of changing the size of the cone sampling window. *Zhang et al. (2015)* reported an average peak density of 168,162 cones/mm$^2$ in 40 eyes although they used a much smaller 5 $\times$ 5 micron sampling window to measure the peak. *Wilk et al. (2017)* reported an average peak density of 145,900 cones/mm$^2$ in 22 eyes using a 37 $\times$ 37 micron sampling window and *Li et al. (2010)* reported an average peak density of 150,412 cones/mm$^2$ in four eyes over a sampling window encompassing 150 cones (approximately 37 micron diameter at the foveal center). All reports of cone densities from adaptive optics studies in living eyes are lower than reports from histology. Two possible reasons for this are (i) the excised tissue in *Curcio et al. (1990)* underwent more shrinkage

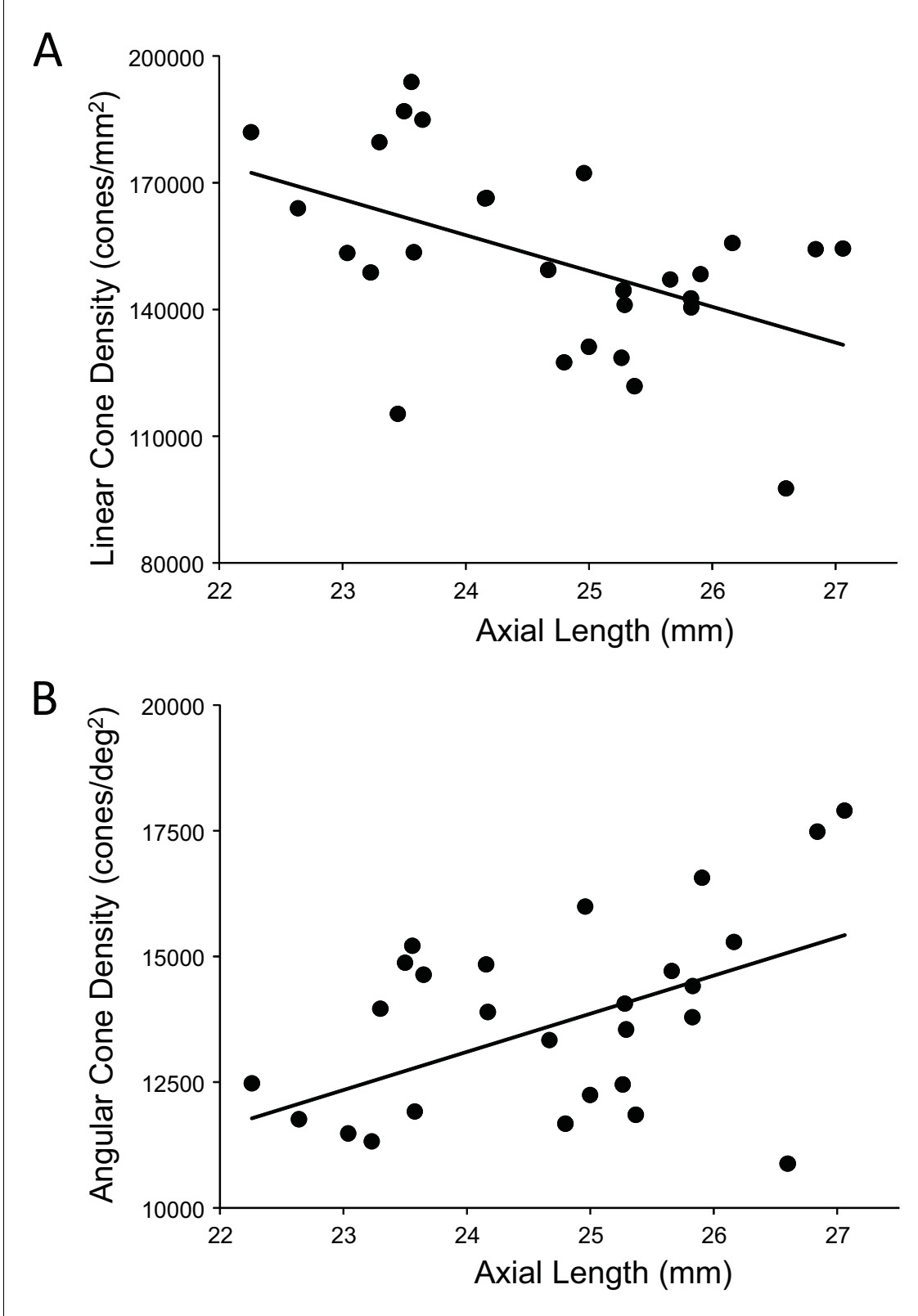

**Figure 6.** The relationship between cone density and axial length shows the same pattern at the PRL as for the peak cone density. The numbers for the trendline in (**A**) are slope: 759; 95% CI: 198 to 1,320; p=0.00999. The numbers for the trendline in (**B**) are: slope = −8,490; 95% CI −14,600 to −2,420; p=0.00795). Axial length accounts for 24%% and 23% of the variance in linear and angular cone density, respectively.

DOI: https://doi.org/10.7554/eLife.47148.018

*Figure 6 continued on next page*

*Figure 6 continued*

The following source data is available for figure 6:

**Source data 1.** Data for plots of cone density as a function of axial length at the PRL.

DOI: https://doi.org/10.7554/eLife.47148.019

than estimated or (ii) the adaptive optics reports are subject to selection bias, where individuals with the highest angular cone densities might have been excluded because the image were less well resolved rendering the cones images too difficult to label with confidence. In our study, we attempted to image 73 eyes from 46 subjects and only succeeded in resolving cones across a sufficiently large region at and around the fovea in 28 of them. The reason the images from 45 eyes were not analyzed was due to poor or inconsistent image quality arising from a number of factors: Images from four eyes (three subjects) were not analyzed because their refractive errors were too high (all above –8D) and we ran into the limits of the deformable mirror's dynamic range. Images from 18 eyes (13 subjects) that were taken early on in the study were not analyzed because the optics of AOSLO were not tuned well enough to resolve foveal cones. Images from four eyes (two subjects) were not analyzed because of uncorrectable image degradation caused by keratoconus and corneal scarring. Images from two eyes (one subject) were not analyzed because of excessive aberrations caused by an orthokeratology refractive correction. The cause of poor or inconsistent image quality among the remaining 17 eyes were varied, including ocular surface dryness, excessive eye motion and small pupils. The average refractive error among these remaining 17 eyes was about the same as the successful eyes.

## Anisotropic density distribution

Like *Curcio et al. (1990)* and *Zhang et al. (2015)* we found steeper drops in cone density in the superior and inferior directions compared to the nasal and temporal directions. Plots of density along the two cardinal directions are shown on *Figure 4—figure supplements 2*.

## PRL displacements

The distance of the PRL from the foveal center for our study (mean 30.4 microns; range 5.7–83.7; n = 28) roughly agrees with those of *Wilk et al. (2017)* (mean 63 microns; range 20–263; n = 22), *Li et al. (2010)* (mean 34 microns; range 3–92; n = 18) and *Putnam et al. (2005)* (mean 17; range 11–23; n = 5). The differences in cone density between the peak and the PRL were small and the trends (*Figures 5* and *6*) persisted at both locations.

## Spatial vision estimates

The cone array imposes the first retinal sampling limit to human spatial vision (*MacLeod et al., 1992*; *Williams, 1985*) and the photoreceptor row-to-row spacing (assuming an hexagonal packing structure) imposes the maximum frequencies that can be relayed to later stages without aliasing. We can compute the sampling limit and estimate the cone center-to-center spacing using the following formulas:

$$Sampling\ Limit = \frac{1}{2}\sqrt{\frac{2}{\sqrt{3}}AngularDensity}$$

$$Cone\ Spacing = Sampling\ Limit^{-1} \times 60 \times \frac{1}{\sqrt{3}}$$

For the densities reported here, the potential spatial frequency resolution limits range from 59.1 to 74.01 cyc/deg (average: 64.9 cyc/deg) at the peak density and 50.4 to 71.9 cyc/deg (average: 62.5 cyc/deg) at the PRL. These correspond to potential acuities ranging from 20/11.9 to 20/8.1 (based on the primary spatial frequency of the three bars of a Snellen E). The cone frequency cutoffs are higher than almost all the interferometric acuity limits reported by *Coletta and Watson (2006)*, even for the emmetropic subjects. The acuities are, however, in the range of those measured from emmetropic subjects after adaptive optics correction by *Rossi et al. (2007)*. The cone center-

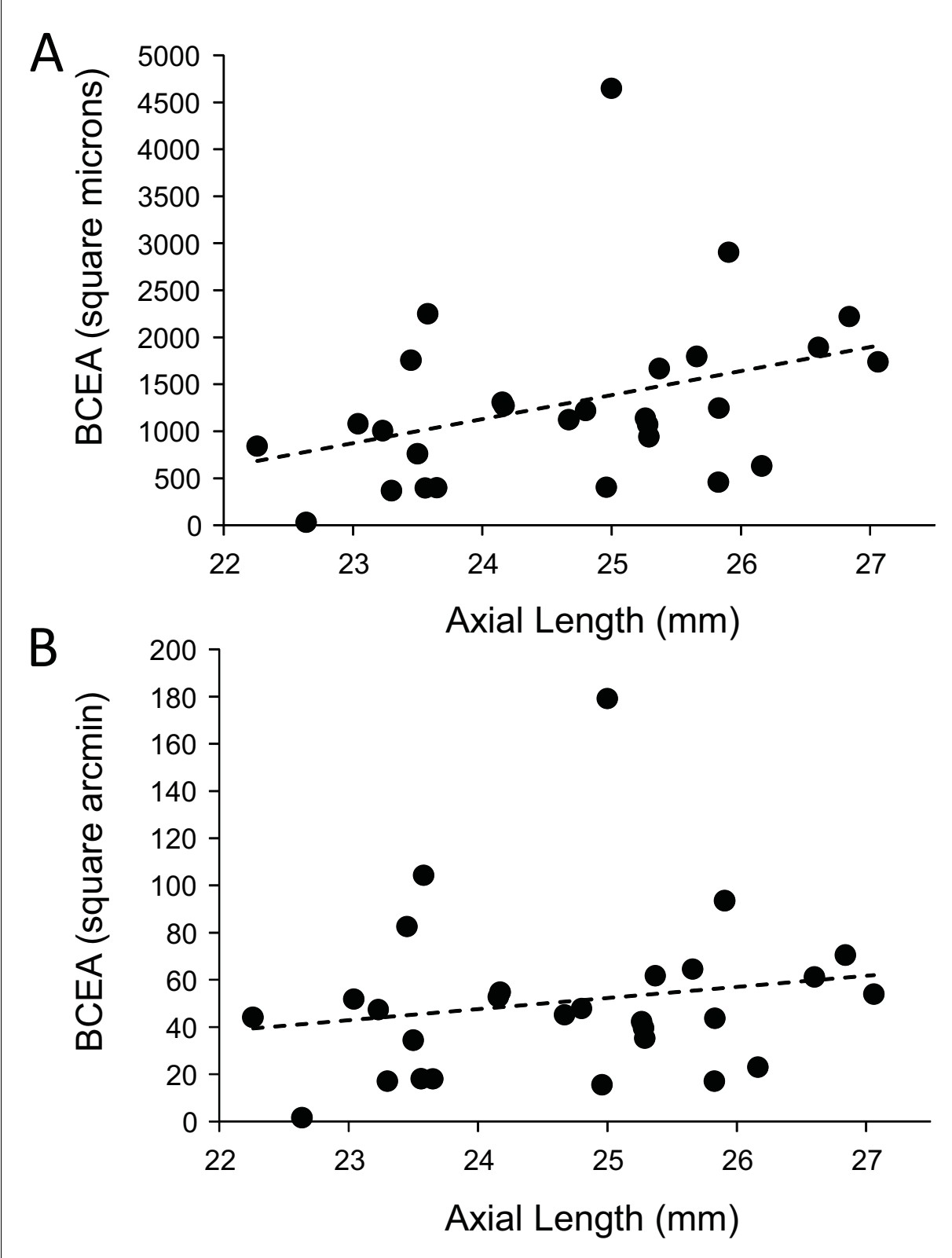

**Figure 7.** Plots of the magnitude of fixational eye movements as a function of axial length. (**A**) The plot of BCEA in linear units (square microns) vs. axial length shows a trend that approaches significance (p=0.0596). (**B**) There is no significant relationship between BCEA in angular units (square arcminutes) and axial length (p=0.364).

DOI: https://doi.org/10.7554/eLife.47148.020

*Figure 7 continued on next page*

*Figure 7 continued*

The following source data is available for figure 7:

**Source data 1.** Data for plots of the magnitude of fixational eye movements as a function of axial length.

DOI: https://doi.org/10.7554/eLife.47148.021

to-center spacing ranges from 0.59 to 0.47 arcminutes at the peak density and from 0.69 to 0.48 arcminutes at the PRL. A direct comparison of foveal structure and function for each of our subjects was not the scope of this study but will be the topic of future investigation.

Measuring structure and function of cone photoreceptors at the foveal center – the most important region of the human retina – has been one of the more challenging endeavors in vision science. Fortunately, the latest generation of adaptive optics ophthalmoscopes are making it easier and are facilitating new discoveries within this retinal region. The pattern of how cone density changes with eye growth lands somewhere between the global expansion and equatorial stretching models. The cone mosaic in longer eyes is expanded at the fovea, but not in proportion to eye length. Despite retinal stretching, myopes generally have a higher angular sampling density in and around the fovea compared to emmetropes. Reports of reduced best-corrected central visual acuity in myopes compared to emmetropes cannot be explained by decreased photoreceptor density caused by retinal stretching during myopic progression.

# Materials and methods

## Foveal imaging

We used our latest generation adaptive optics scanning laser ophthalmoscope (AOSLO) for foveal imaging. The system used a mirror-based, out-of-plane optical design (*Dubra et al., 2011*), and employed a deformable mirror with a continuous membrane surface and shaped with 97 actuators (DM97, ALPAO, Montbonnot-Saint-Martin, France). The system scans multiple wavelengths simultaneously. Each wavelength was drawn from the same broadband supercontinuum source (SuperK EXTREME, NKT Photonics, Birkerod, Denmark) using a custom-built fiber coupler. Wave aberrations were measured with a custom-built Shack Hartmann wavefront sensor using the 940 nm channel. Images were recorded using the 680 nm channel. $512 \times 512$ pixel videos were recorded over a $0.9 \times 0.9$ degree square field for an average sampling resolution of 9.48 pixels per arcminute. Eye alignment and head stabilization was achieved by using either a bite bar or a chin rest with temple pads. At least one 10 s video was recorded at the fovea and at eight more locations where the subjects were instructed to fixate on the corners and sides of the raster, to image an entire foveal region spanning about $1.8 \times 1.8$ degrees. In order to ensure the best possible focus of the foveal cones, multiple videos were taken over a range of 0.05 D defocus steps to find the sharpest foveal cones. Focus steps were generated by adding a focus shape onto the deformable mirror. Online stabilization and registration algorithms were used to facilitate rapid feedback on the image quality.

## Locating the Preferred Retinal Locus of Fixation (PRL)

Steady fixation was achieved at the fovea center by having the subjects fixate on a dark, circular, blinking dot with a diameter of 3.16 arcminutes (30 pixels) in the center of the AOSLO scanning raster. The fixation target was generated by modulating the same 680 nm scanning beam used for imaging and, as such, the target's location was encoded directly into each frame of the video (*Poonja et al., 2005*). A scatter plot of the positions of the blinking dot relative to the retina was generated and was fit with a bivariate ellipse using free online MATLAB scripts downloaded from http://www.visiondummy.com/wp-content/uploads/2014/04/error_ellipse.m. The bivariate contour ellipse area (BCEA), which is the area of the best-fitting ellipse encompassing 68% of the points in the scatterplot (*Castet and Crossland, 2012*) was used to quantify the fixation stability (*Figure 7*) and the exact location of the PRL within the imaged cone mosaic (*Table 2*, *Figure 3—figure supplements 1* and *2*).

## Image processing and analysis

High quality images were generated from the recorded videos offline using custom software (MAT-LAB, The MathWorks, Inc, Natick, MA) to measure and correct for distortions caused by eye movements (*Stevenson and Roorda, 2005*). An open source version of the code is posted at https://github.com/lowvisionresearch/ReVAS (*Agaoglu et al., 2018*; copy archived at https://github.com/elifesciences-publications/ReVAS). Poor-quality frames were manually excluded and registered frames were averaged into a single high signal-to-noise image. The processed images were stitched together (Photoshop; Adobe Systems, Inc, Mountain View, CA) to create an approximately 1.8-degree montage of the foveal cone mosaic.

We used custom software to identify and label individual cones in the AO retinal images. The program allows the user to select a region of interest and manually add and delete cone labels. A combination of both manual and automated methods (*Li and Roorda, 2007*) were used to identify cone locations as the current version of the program does not adequately recognize cones in the foveal center where they are dim and smaller (*Li et al., 2010*). All the cone coordinates were selected and reviewed by two of the authors. In some cases cones were too dim to be seen but there was only a gap in the mosaic (*Bruce et al., 2015*). If a space that might have been occupied by a cone was dim or dark, we would assume it was a cone and mark its location. We rationalize this for two reasons: First, if there is a gap in the mosaic, then it is likely that a cell is occupying that space, otherwise the adjacent cells would migrate to fill it in (*Scoles et al., 2014*). Second, in our experience and of others (*Pallikaris et al., 2003*), cones that appear dark in one visit, can often appear bright in the next. In other cases (uncommon) the contrast was low in some regions or there were interference artifacts in the images (*Meadway and Sincich, 2018*; *Putnam et al., 2010*), making the cone locations slightly ambiguous. In these instances, we made manual cone selections based on the assumption that the cones were all similar in size and close-packed into a nearly hexagonal array (*Curcio et al., 1990*).

Continuous density maps were generated by computing cone density within a circular cone sampling window of 10 arcminutes in diameter around every pixel location across the image. All cones whose centers fell on or within the boundary of the sampling window were counted. The size of the cone sampling windows was chosen for several reasons. First, it encompasses the typical range of eye motion during a typical fixation task. Second, the area was large enough to generate smooth maps, but small enough to resolve local changes. Smaller cone sampling windows showed slightly higher densities (see Discussion) but were also associated with more variability in the identification of the location of peak cone density. Changes in density with eccentricity were generated by computing the density in five-arcminute annuli surrounding the point of peak cone density. For linear density measures we used annuli with 25-micron widths.

## Retinal Magnification Factor calculation

The exact angular dimensions of the AOSLO images were computed by imaging a calibrated model eye in the AOSLO system, but the conversion to linear dimensions on the retinal image requires additional measurements, since the dimensions of each eye governs the actual size of the image on its retina. The conversion from visual angle to retinal distance requires a measurement of the axial length of the eye and an estimation of the location of the secondary nodal point. We did paraxial ray tracing, described by *Li et al. (2010)*, in a four-surface schematic eye model to estimate the location of the secondary nodal point. The corneal first surface radius of curvature, the anterior chamber depth and the axial length were measured for each subject with an IOL Master (Zeiss Meditec, Dublin, CA). The radius of the curvature of the back surface of the cornea was computed as 88.31% of the front surface as per the Gullstrand eye model (*Bennett et al., 1994*). The indices of refraction of the media and the radii of curvature of the front and back lens surface were taken from the Gullstrand schematic eye (*Vojniković and Tamajo, 2013*). Once determined, retinal image size is related to visual angle by the equation:

$$I = \tan(1°)(x - AN')\theta$$

Where *I* is retinal image size, *x* is axial length, *AN'* is the distance from the corneal apex to the eye's second nodal point, and θ is the visual angle. As can be seen in *Table 2*, myopic eyes, which generally have longer focal lengths, have proportionally larger retinal images.

Other methods to compute the retinal magnification factor (RMF) might have been used. We did a paraxial ray trace to find the secondary nodal point using lens parameters from a 6-surface Gullstrand eye model, and found that the RMFs were essentially identical. When we computed the RMF using an equation proposed by *Bennett et al. (1994)* we found that the RMF was overestimated for hyperopic eyes and underestimated for myopic eyes. This discrepancy arises from the fact that Bennett's equation is based on the assumption that the focal point of the eye's optics coincides with the retina, which it does not (*Li et al., 2010*). Despite this known error, when we redid the analysis using RMFs computed using the *Bennett et al. (1994)* equation, we still found a significant trend of reducing linear density with axial length, but only at the position of peak cone density. Of course, all the angular density calculations are unaffected by differences in the RMF.

## Statistical analysis

Given the trends of increased angular density as a function of axial length that *Li et al. (2010)* observed at the location closest to the fovea (slope = 531 cones/deg$^2$ for each mm increase in axial length; standard deviation of the regression errors = 1377 cones/deg$^2$), we estimated that data from approximately 32 eyes, evenly distributed across a range of axial lengths would be sufficient to show if there was a true effect at the fovea. The targeted number was computed using methods outlined by *Dupont and Plummer (1998)* implemented using free online software (Power and Sample Size Program Version 3.0, January 2009, downloaded from http://biostat.mc.vanderbilt.edu/wiki/Main/PowerSampleSize) with type one error probability of 0.05 and a power of 0.95.

All data collected in this study were analyzed using simple linear regression models in Excel. p-values for all linear regressions are reported and linear trendlines with p-values less than 0.05 are plotted as solid lines and p-values greater than 0.05 as dashed lines.

In this study, we included both the right and left eyes of some of the subjects. When we reran the statistics with a single eye from each subject (n = 16), the results were essentially the same and the same conclusions could be drawn.

## Additional information

### Competing interests
Austin Roorda: has a patent (USPTO#7118216) assigned to the University of Houston and the University of Rochester which is currently licensed to Boston Micromachines Corp (Watertown, MA, USA). Both he and the company stand to gain financially from the publication of these results. No other authors have competing interests. The other authors declare that no competing interests exist.

### Funding

| Funder | Grant reference number | Author |
|---|---|---|
| National Eye Institute | R01EY023591 | Nicolas Bensaid<br>Pavan Tiruveedhula<br>Jianqiang Ma<br>Austin Roorda |
| National Eye Institute | K08EY025010 | Sowmya Ravikumar |
| National Eye Institute | T35EY007139 | Yiyi Wang |
| National Eye Institute | P30EY003176 | Austin Roorda |

The funders had no role in study design, data collection and interpretation, or the decision to submit the work for publication.

### Author contributions
Yiyi Wang, Data curation, Formal analysis, Investigation, Visualization, Writing—original draft, Writing—review and editing; Nicolas Bensaid, Pavan Tiruveedhula, Jianqiang Ma, Software, Significant engineering support; Sowmya Ravikumar, Conceptualization, Data curation, Investigation, Writing—original draft; Austin Roorda, Conceptualization, Data curation, Software, Formal analysis,

Supervision, Funding acquisition, Validation, Investigation, Visualization, Methodology, Writing— original draft, Project administration, Writing—review and editing

## Author ORCIDs
Austin Roorda https://orcid.org/0000-0002-3785-0848

## Ethics

Human subjects: The work on human subjects was reviewed and approved by the University of California, Berkeley Institutional Review Board and is described in protocol # 2010-08-2006. All subjects provided informed consent prior to any experimental procedures.

## Decision letter and Author response
Decision letter https://doi.org/10.7554/eLife.47148.030
Author response https://doi.org/10.7554/eLife.47148.031

# Additional files

## Supplementary files
• Transparent reporting form
DOI: https://doi.org/10.7554/eLife.47148.022

## Data availability
The following data are available for download at the Dryad Digital Repository: https://dx.doi.org/10.5061/dryad.nh0fp1b. 1. All original images and cone locations. 2. A table of scaling parameters (pixels per degree and retinal magnification factors) for each image. 3. A MATLAB script that can be used to plot cone locations on the original image.

The following dataset was generated:

| Author(s) | Year | Dataset title | Dataset URL | Database and Identifier |
|---|---|---|---|---|
| Wang Y, Bensaid N, Tiruveedhula P, Ma J, Ravikumar S, Roorda A | 2019 | Data from: Human Foveal Cone Photoreceptor Topography and its Dependence on Eye Length | https://dx.doi.org/10.5061/dryad.nh0fp1b | Dryad Digital Repository, 10.5061/dryad.nh0fp1b |

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
