## [Decision Letter]

Thank you for submitting your manuscript "Human foveal cone photoreceptor topography and its dependence on eye length" for consideration by *eLife*. Your article has been reviewed by a Senior Editor, a Reviewing Editor, and two reviewers.

As you will see, both reviewers were impressed with the importance and novelty of your work. I am including the reviews at the end of this letter, as there are a variety of specific and useful comments in them. As you will see, the reviewers' comments relate primarily to data analysis and interpretation.

We look forward to receiving the revised version of your manuscript.

*Reviewer #1*

Wang et al. tackle an issue of fundamental importance in understanding mechanisms underlying human myopia as well as the basic processing of foveal visual signals. In general, I find the data to be high quality and the presentation to be quite strong. That said, there are some "moderate" issues that I think are important regarding rigor and transparency. First, the repeatability of foveal cone density measures has not been established. While there are automated algorithms assessing parafoveal cone mosaic, and reliability and repeatability has been demonstrated at those locations, I think the foveal cone mosaic is sufficiently different from an analytical perspective to warrant repeated measures. Similarly, the choice of model eye to convert angular to linear could be discussed more and ruled out as a contributing factor to your results (given there may be other models for converting to/from linear and angular density). Lastly, statistically, I think there may be some issues with using two eyes from some subjects but only one from others. Many ocular parameters show interocular symmetry, so the non-independence of some of the data should be reviewed by someone with more statistical expertise than I have.

In addition, the choice of sampling window was not discussed. To the one extreme, one could take the reciprocal of the area of a single cone at the very center and have an overestimate of "density". Or, one could sample over the entire retina and have an underestimate of peak density. I think some justification is warranted, 25 microns may be as good as any, but if you used either a different fixed window, or picked some number of cones (100?) and increased the window until 100 cones were found may reinforce the data. A smaller point, was the sampling window "bound" or did you just count cells in a given window? It might be easy to re-run the coordinates with different sampling approaches to see how dependent the relationships you report are or are not dependent on this.

Also, as shown by Cooper et al., density is one of the more "sensitive" metrics, meaning that just a couple cones in error during identification can result in big changes in density. Spacing is more robust, where a few missed cells will not affect the local spacing estimate. Did you look at NND or ICD in μm and arcmin? Do you find the same differences?

Finally, there are a couple of instances in Table 2 where the density at the PRL is actually higher than that reported as the "peak foveal cone density" (20124, OD; 20170, OS). This is likely an issue related to the sampling window, but worth looking at as it is counter-intuitive.

*Reviewer #2*

I am most impressed by this paper. It is exceptionally well done in my view, and sets the standard for work of this sort. Previous work in this area of research is carefully reviewed and critiqued, and it seems to me that every paper related to the measurements made in the present paper is included, although admittedly I am not all that familiar with the research addressing the issue of foveal cone density as a function of axial length.

I found the paper well written, clear and compelling. The figures are well done and convincing. I have no issues with the findings, techniques or conclusions, and the discussion and supplementary data were useful.

My one question and suggestion, for the discussion perhaps, is why the preferred retinal locus for fixation (PRL) is displaced in many eyes from the area of peak cone density (Figure 3 and Figure 3—figure supplements 1 and 2). This is not a new finding as noted, but what is the present thinking regarding this? Could it be related to differences to foveal cone wiring as suggested in the Discussion for causes of myopic visual deficits? Is there a hint of this in Table 2 in which those subjects with longer axial lengths seem to have longer PRL distances from the foveal center? Some discussion of this would be useful.

---

## [Author Response]

Reviewer #1Wang et al. tackle an issue of fundamental importance in understanding mechanisms underlying human myopia as well as the basic processing of foveal visual signals. In general, I find the data to be high quality and the presentation to be quite strong. That said, there are some "moderate" issues that I think are important regarding rigor and transparency. First, the repeatability of foveal cone density measures has not been established. While there are automated algorithms assessing parafoveal cone mosaic, and reliability and repeatability has been demonstrated at those locations, I think the foveal cone mosaic is sufficiently different from an analytical perspective to warrant repeated measures.

The reviewer is correct that reliability and repeatability of cone density measurements have never been reported for foveal cones. However, we used two experienced image analysts (two of the authors) working in tandem to carefully identify every cone in the mosaic. The criteria for labeling cones was described explicitly in the manuscript. The effort took countless hours and only images and data for which both analysts were confident were used for further analysis.

To undertake a reliability/repeatability study for this analysis would require training two or more new image analysts to the same level of the current analysts. Even if funds and personnel were available, the time delays incurred by such an effort would be prohibitive.

Nevertheless, we do appreciate the reviewers point and to that end we have made all the images and cone locations available for download to any other interested group. We envision that these data could be used for an independent analysis of repeatability and reliability or our careful selections could be used to train AI-based approaches to count cones.

Similarly, the choice of model eye to convert angular to linear could be discussed more and ruled out as a contributing factor to your results (given there may be other models for converting to/from linear and angular density).

Our conversion between visual angle and linear distance on the retina was done using the maximum biometric data available to us from a Zeiss IOLMaster device. We employed paraxial ray tracing methods that have been described and discussed in a previous paper from our lab (Li et al., 2010).

The primary aim of the ray trace is to find the secondary nodal point of the eye. The secondary nodal point is one of the cardinal points of a thick lens optical system. Nodal points have the important property that the visual angle of object subtended from the primary nodal point matches the angle of its retinal image subtended by the secondary nodal point. If one knows the distance from the secondary nodal point to the retina, then the actual size of the image of any object of known visual angle can be computed using simple geometric calculations. Since our biometry was limited to corneal curvature, anterior chamber depth and axial length, we had to make some approximations to some of the variables in the ray trace, specifically the thickness of the cornea, the power of the second surface of the cornea relative to the first surface, and the power and thickness of the crystalline lens. For these, we used parameters from the Gullstrand model eye for a two-surface lens.

As per the reviewer’s suggestion, we recomputed the retinal magnification factor (RMF, in microns per degree) for a four-surface lens model and obtained nearly identical results. Any further analysis with these RMFs would be nearly indistinguishable from what is already in the manuscript so we did not include them in this response. The results are shown on Author response table 1.

**Author response table 1. resptable1:** Retinal Magnification Factors (microns/degree) computed three different ways.

Subject	Eye	RMF (2-surface lens)	RMF (4-surface lens)	RMF (Bennett et.al., 1994)
20165	L	261.73	263.60	266.95
	R	267.81	269.64	271.91
20177	L	273.56	275.04	277.13
	R	275.86	277.29	279.61
10003	L	278.75	280.19	280.53
	R	282.02	283.46	283.14
20176	L	276.44	278.03	282.49
	R	278.54	280.11	284.19
20172	L	280.15	281.93	283.92
	R	281.29	283.09	285.10
20147	R	298.76	299.90	291.76
	L	288.94	290.74	291.89
20124	L	298.86	300.71	298.42
	R	309.82	311.68	306.52
20174	L	302.60	304.23	300.12
	R	311.83	313.46	307.56
20173	R	304.58	306.24	302.21
20170	R	305.52	307.31	302.73
	L	316.23	318.13	311.35
20138	R	311.19	312.58	306.13
	L	311.93	313.33	306.39
20114	R	310.96	312.19	313.57
	L	313.35	314.53	317.88
20160	R	320.19	321.82	313.57
20143	R	334.10	335.25	314.62
20158	R	333.78	335.36	323.63
20163	R	336.64	338.26	326.76
	L	340.48	342.10	329.63

The table also includes the RMF computed using a popular method proposed by Bennett et al., 1994 (the paper has been cited over 350 times according to Google Scholar). The formula is

RMF = 1000*tan(1/1.336)*(AL-1.82)

where AL is the axial length in mm, 1.82 is the distance in mm from the corneal vertex to the second principal plane in the Gullstrand model eye, and 1.336 refers to the index of refraction of the vitreous.

Incidentally, we used the Bennett equation in an earlier paper (Rossi et al., 2007), but we’ve since learned that it produces incorrect results. As stated in Li et al. (2010), the Bennett formula underestimates the RMF for myopes and overestimates the RMF for hyperopes since “this method assumes that the retina and the eye’s back focal plane coincide, which is not the case in myopia [or hyperopia]”. Despite underestimating the RMF for high myopia, a significant decrease in linear density with eye length is still observed, although only at the location of peak foveal density. The plots are shown on Author response image 1. Note that we are including these plots to show an extreme case – we do not recommend anyone use this formula, unless they are sure they are working with emmetropic eyes.

We have expanded the discussion of this in a new paragraph at the end of Materials and methods: Retinal Magnification Factor section of the revised manuscript.

**Author response image 1. respfig1:** Plots of linear cone density at different retinal locations using retinal magnification factors computed using a formula from Bennett et.al., 1994. Even though the retinal image size for myopes is underestimated for myopes and overestimated for hyperopia, there is still a significant drop in density with increasing axial length at the location of peak density.

Lastly, statistically, I think there may be some issues with using two eyes from some subjects but only one from others. Many ocular parameters show interocular symmetry, so the non-independence of some of the data should be reviewed by someone with more statistical expertise than I have.

We agree that there is symmetry in cone density between the two eyes, but we opted to include them to maximize the number of eyes in the study. We repeated all the statistics with one eye per subjects (16 subjects total) and we got essentially the same result. Author response image 2 shows trends that are similar to those shown in Figure 5 but with one eye from 16 subjects.

**Author response image 2. respfig2:** Changes in linear and angular cone density with axial length over a range of distances from the location of peak density. The results for one eye are essentially the same as that reported in the paper.

We discuss this limitation in the revised manuscript in the final paragraph of the section Materials and methods: Statistical Analysis in the revised manuscript.

In addition, the choice of sampling window was not discussed. To the one extreme, one could take the reciprocal of the area of a single cone at the very center and have an overestimate of "density". Or, one could sample over the entire retina and have an underestimate of peak density. I think some justification is warranted, 25 microns may be as good as any, but if you used either a different fixed window, or picked some number of cones (100?) and increased the window until 100 cones were found may reinforce the data. A smaller point, was the sampling window "bound" or did you just count cells in a given window? It might be easy to re-run the coordinates with different sampling approaches to see how dependent the relationships you report are or are not dependent on this.

The sampling window is an important parameter governing peak cone density measurements which is why we stated the sampling windows along with cone densities from all published studies in Discussion: Comparisons with Previous Studies. Our choice of 10 arcmins or 50 microns diameter was chosen primarily because it encompassed the average range of fixation stability. In our cohort, the average standard deviation of eye movement during fixation was 3.94 arcminutes. As described in Discussion: Comparisons with Previous Studies, we reduced the sampling window to 7.5 arcminutes in the discussion to make a more direct comparison with histological data reported by Curcio et al., 1990.

The density analysis included all cones whose centers fell within the cone sampling window. This is now stated clearly in the last paragraph of the Materials and methods: Image Processing and Analysis section.

On the suggestion of the reviewer we performed an analysis of the effect of the sampling window size. We computed peak cone density and location of peak cone density for all 28 eyes for a range of cone sampling windows. The results are shown in Author response image 3. The plots suggest that cone sampling windows of less than 10 minutes of arc may add more variability to the cone density estimates and fovea center identification. We added some more discussion about this in the final paragraph of the Materials and methods: Image Processing and Analysis section.

**Author response image 3. respfig3:** The right plot shows that the peak foveal cone density increases as the cone sampling window is decreased. Note that the increase with reducing sampling window is linear until about 10 arcminutes. The left plot shows that the variability in the location of the peak foveal density remains consistent (within about 1.5 arcminutes of the mean) with cone sampling windows of 10 arcminutes or greater.

Also, as shown by Cooper et al., density is one of the more "sensitive" metrics, meaning that just a couple cones in error during identification can result in big changes in density. Spacing is more robust, where a few missed cells will not affect the local spacing estimate. Did you look at NND or ICD in μm and arcmin? Do you find the same differences?

We chose not to compute any metrics other than density. In our case, we were confident that all cones were labeled. Under such conditions, density is the most informative metric to report.

With regard to sensitivity, the number of cones within the 10 arcmin diameter cone sampling window at the foveal peak ranged from 264 to 414. A couple of extra and/or missing cone labels would lead to errors of <1%. With good images and careful identification of cones according to our criteria stated in the Materials and methods, density is actually a very sensitive metric.

Given the close-packed nature of the foveal cones, we opted to compute row-to-row spacing of cones directly from the density estimates.

Finally, there are a couple of instances in Table 2 where the density at the PRL is actually higher than that reported as the "peak foveal cone density" (20124, OD; 20170, OS). This is likely an issue related to the sampling window, but worth looking at as it is counter-intuitive.

We thank the reviewer for spotting our oversight of this discrepancy. After reviewing our analysis code (Matlab) we discovered some rounding errors, specifically in our conversions from distances to pixels, that led to a series of small errors in our densities, including the ones that led to the two discrepancies spotted by this reviewer. We have since redone all the analyses for all eyes and have generated an entirely new set of density values for which we are much more confident. The new peak densities are slightly higher than originally reported by an average of 1.2% and the densities at the PRL are higher by 0.3%. Because of the new density values, we had to rerun all the statistics, corrected all numbers in the final four columns of Table 2, made small correction to many numbers throughout the manuscript, made changes to Figures 4, 5, 6, 7, and the three supplementary figures. These changes have not affected the primary findings or conclusions of this paper in any way.

Reviewer #2I am most impressed by this paper. It is exceptionally well done in my view, and sets the standard for work of this sort. Previous work in this area of research is carefully reviewed and critiqued, and it seems to me that every paper related to the measurements made in the present paper is included, although admittedly I am not all that familiar with the research addressing the issue of foveal cone density as a function of axial length.I found the paper well written, clear and compelling. The figures are well done and convincing. I have no issues with the findings, techniques or conclusions, and the discussion and supplementary data were useful.My one question and suggestion, for the discussion perhaps, is why the preferred retinal locus for fixation (PRL) is displaced in many eyes from the area of peak cone density (Figure 3 and Figure 3—figure supplement 1 and 2). This is not a new finding as noted, but what is the present thinking regarding this? Could it be related to differences to foveal cone wiring as suggested in the Discussion for causes of myopic visual deficits? Is there a hint of this in Table 2 in which those subjects with longer axial lengths seem to have longer PRL distances from the foveal center? Some discussion of this would be useful.

The displacement between the PRL and the location of peak cone density is of interest to us but we do not want to offer any more discussion in the manuscript than we have already since it would all be speculative. Also, despite what the reviewer might see, there is no correlation between axial length and PRL displacement in both angular and linear distance.

But here are some opinions:

The differences in cone density between the PRL and peak cone density is not visually significant, as the sampling resolution at either location is generally better than typical acuity performance, and the cones oversample the retinal image at both locations. This is not stated explicitly but is implicit from the discussion.

At this time, we have not performed sufficient functional measurements to reveal what, if anything, is optimized at the PRL vs. the position of peak cone density.

Our working opinion is that a choice for the PRL is made during development, at a time when biophysical factors that cause reshaping of the foveal region are still taking place. The fact that the PRL, peak cone density, foveal avascular zone, and base of the foveal pit are so well-aligned in the normally-developed adult eye means that the right choice was generally made.